# Interplay Between Ocular Ischemia and Glaucoma: An Update

**DOI:** 10.3390/ijms252212400

**Published:** 2024-11-19

**Authors:** Valeria Coviltir, Miruna Gabriela Burcel, George Baltă, Maria Cristina Marinescu

**Affiliations:** 1Ophthalmology Discipline, Carol Davila University of Medicine and Pharmacy, 020021 Bucharest, Romania; valeriacoviltir@yahoo.com; 2Clinical Hospital for Ophthalmological Emergencies, 010464 Bucharest, Romania; george.balta@drd.umfcd.ro; 3Faculty of Medicine, Transilvania University of Braşov, 500019 Braşov, Romania; 4Brasov County Emergency Clinical Hospital, 500326 Braşov, Romania; 5Doctoral School, Carol Davila University of Medicine and Pharmacy, 020021 Bucharest, Romania; 6Medical Physiology Discipline, Carol Davila University of Medicine and Pharmacy, 020021 Bucharest, Romania; maria.marinescu@drd.umfcd.ro

**Keywords:** glaucoma, retinal ischemia, vascular dysregulation

## Abstract

Glaucoma is a main cause of irreversible blindness worldwide, with a high impact on productivity and quality of life. The mechanical and ischemic theories are currently the most recognized pathophysiological pathways that explain the neurodegeneration of retinal nerve fibers in glaucoma. In this narrative review, aspects of ischemia in glaucoma are discussed, including vascular dysregulation, retinal ischemia signaling pathways, roles of vascular endothelial growth factors, and future research and therapeutic directions. In conclusion, a better understanding of the ischemic processes in glaucoma may lead to innovative treatment options and improved management and follow-up of our patients.

## 1. Background

Ischemia is a process that consists of insufficient blood flow to a certain area due to an obstruction of blood vessels. In ocular tissues, the obstruction is at the level of capillary vessels, causing non-perfusion, which leads to neural dysfunction, and subsequently, cell death [1]. This is accompanied by a self-reinforcing destructive cascade involving glial cell stimulation and the production of cytotoxic mediators (such as cytokines). Ocular diseases in which ischemia was found as a driving force include diabetic retinopathy and the subject of the present review, glaucoma [1].

Glaucoma is a neurodegenerative disease that affects around 3.5% of the population worldwide, and it is reported that 5.7 million people are visually impaired and 3.1 million people are blind due to its damage [2,3]. Predictions estimate that, by the year 2040, 111.8 people will have glaucoma [2,3].

Besides the individual health impact, glaucoma has a significant economic impact as well. It is estimated that in Europe, the yearly cost of glaucoma management is between 455 and 969 euros for each patient, while in the USA, it has reached a global cost of 2.5 billion dollars yearly. Glaucoma is generally caused by progressive damage to the retinal ganglion cells (RGCs) and optic nerve fibers, ultimately leading to visual field constriction, in which increased intraocular pressure (IOP) is the main risk factor [4,5,6,7]. Elevated intraocular pressure is the main risk factor in glaucoma, and the only one currently addressable through therapy—either medical or surgical. However, in certain cases, the glaucomatous RGC and optic nerve degeneration progresses further.

So, improving knowledge of the ischemia processes may contribute to developing novel treatment options.

While glaucoma is a complex disease and the entire pathological process is not fully understood, two main physiopathological theories have been proposed to explain glaucomatous neuropathy: the vascular and the mechanical theories [5,6,7].

Also, glaucoma has been studied in the context of neurodegenerative disorders, such as Alzheimer’s disease [8]. There is increasing evidence that common neurodegenerative diseases, such as Alzheimer’s disease and vascular dementia, share a common pathophysiologic pathway. Of renewed interest are the autoimmunological and inflammatory responses that influence RGC loss [8,9,10].

## 2. Hallmark Theories of Glaucoma Pathogenesis

The pathophysiology of glaucoma involves complex interactions between various cell types in the eye such as RGCs, astrocytes, microglia, and endothelial cells. The exact mechanisms underlying optic nerve damage and RGC death in glaucoma are not completely understood, but multiple theories have been proposed. The most studied is the mechanical one; however, there are also other factors cited that contribute to structural changes in the optic nerve head in glaucoma including oxidative stress, inflammation, neurotrophic factor deficiency, and mitochondrial dysfunction [11].

The mechanical theory hypothesizes that mechanical pressure on the optic disc and lamina cribrosa distorts the fovea and consequently compresses RGC axons, as well as blocks axoplasmic transport, resulting in the activation of apoptotic and necrotic pathways [7]. Nowadays, the only therapeutically addressable risk factor is elevated IOP, either medically or surgically [12]. Another pivotal theory is the ischemic one, of which crucial factors are illustrated in Figure 1 and detailed in the paragraphs below.

Some authors have explored evidence suggesting that glaucoma may not be a primary disease of the eye, but rather an ocular manifestation of some type of systemic dysfunction [10,13]. The vascular theory suggests that glaucoma is a consequence of insufficient intraocular blood flow. The abnormal ocular perfusion and subsequent ischemia cause retinal ganglion cell death [13,14].

The ocular perfusion pressure (OPP) is the difference between arterial and venous blood pressure. It is known that in the eye, the venous pressure almost equals IOP, thus the OPP can be estimated as the difference between the arterial pressure and IOP. Therefore, anomalies of arterial pressure, either hypertension or hypotension, are thought by some to contribute to the development of glaucoma [12].

Regarding hypertension, it is still being debated whether elevated blood pressure is associated with the development of glaucoma, as there are studies that failed to find this association [15,16]. Hypertension is directly linked to the elevated glaucoma risk, due to an increase in ciliary blood flow and, subsequently, increased aqueous humor production, while high episcleral venous pressure leads to a decrease in humor outflow [15].

On the other hand, multiple population-based studies show that low systolic and diastolic blood pressures are associated with glaucoma progression. Antihypertensive treatment, which leads to low diastolic pressure, was linked with glaucoma progression [10]. Normal-tension glaucoma and arterial hypotension are related in particular, as systemic disease decreases ocular perfusion pressure, which leads to ischemia in optic nerve structures and glaucomatous damage [17].

Vascular dysfunction, and thus, alterations in autoregulation in response to perfusion pressure changes, may cause an unstable blood supply and, as a result, mild but repeated ischemic injury to the optic nerve [18,19]. Insufficient blood flow to retinal structures may be followed by hypoxia, cellular nutrient deficiency, and inefficient waste removal, leading to cellular apoptosis. Alterations in autoregulation in glaucoma appear to involve vascular endothelial dysfunction [20,21].

Retinal ischemia is a process involved in numerous disorders, including glaucoma. As the up-regulation of microglia and the macrophage recruitment was identified in many other neurological diseases, it was extensively studied.

It is known that important vasoactive factors for the retinal blood vessels include Endothelin-1 (ET-1) and nitric oxide (NO), and influence autoregulation of choroidal blood flow. Nitric oxide, a vasodilator molecule, is released by vascular endothelial cells and supports neuroprotection of the retinal nerve fiber layer (RNFL) through the autoregulation of the retinal blood flow [7].

Some studies showed elevated basal nitric oxide synthase (NOS) activity in Primary open-angle glaucoma (POAG), suggesting a compensatory mechanism to ensure adequate ocular blood flow (OBF). Therefore, compared to healthy subjects, choroidal and optic nerve blood flow was less affected in POAG if NOS was blocked systemically [10].

On the other hand, Endothelin-1 acts as a very potent vasoconstrictor, and elevated serum levels are associated with an increased risk of glaucomatous neuropathy [22]. Some studies on animal models revealed that administration of ET-1 near the optic nerve induced retinal vasoconstriction, and consequently ischemia, resulting in regional RGC loss and glaucoma progression [22]. However, besides the vascular actions with potentially damaging effects on the retina, this peptide also promotes the neurotoxic effects of glutamate [23,24,25], which is a major neurotransmitter of the retina, and produces significant neurodegeneration on glial cells and neurons by decreasing cell viability, lowering axonal transport, and through higher astrogliosis.

These non-ischemic effects, which contribute to the glaucoma pathophysiology, explain the need for therapeutic options not only aimed at lowering the IOP or, by other means, increasing perfusion [22].

Nordahl and colleagues investigated the way the retinal function evolved following an intravitreal ET-1 injection (used experimentally as a model of acute retinal ischemia)—using electroretinography (ERG). To develop an animal model of retinal ischemia, they attempted the engineering of a transgene ET-1 rat (with unilateral, chronic high intraocular ET-1). Their investigations revealed that particular transgenic animal models undergo similar ERG changes to those following ET-1 injections, and are similar to those in other retinal ischemia models and glaucoma. Gene expression analysis revealed that the response to increased ET-1 involved the genes for the ET-1 receptor and an up-regulation of the calcitonin gene-related protein (CGRP), which acts as a vasodilator [22].

Caveolin-1 (Cav-1) is another molecule involved in vascular dysregulation, being tightly linked to the pathways involved in NO synthesis in the endothelium (eNOS—endothelial nitric oxide synthase). eNOS is normally expressed by vascular endothelial cells and leads to NO and its vasodilatory effect, overall promoting endothelial health [23,26]. Cav-1 is in direct competition with calmodulin (in this case, an eNOS activator) for the binding site of eNOS [23,26,27]. Chronic excessive up-regulation of eNOS is associated with a Cav-1 decrease, and a lower NO is associated with vascular dysfunction and, in terms of systemic health, cardiovascular mortality [28]. This association between Cav-1 deficiency and vascular dysregulation may exacerbate optic nerve head injury, especially in the presence of the additional stress of high IOP. To summarize, enhancing Cav-1 expression and activity may prove to be a valuable glaucoma therapeutic target, as it is a molecule highly involved in several glaucoma mechanisms [26].

## 3. Interplay Between Ischemia and Oxidative Stress Dysregulation

The retina is one of the most metabolically active tissues in the body and has the highest oxygen demand. It is estimated that in humans, oxygen extraction is approximately 2.33 μL of oxygen per minute, or 1.42 mL per minute per 100 g of retinal tissue, higher than in other tissues [29]. As the energy consumption of the retina is of a similar value to that of the brain, and the retinal vasculature is very sensitive to oxygen availability, it requires an efficient mechanism to maintain homeostasis [30]. As such, the normal functioning of retinal neurons can be affected if nutrients and oxygen are not supplied according to the high tissular demands [30].

As retinal ganglion cells are rich in mitochondria, they have an exceedingly active metabolism and are particularly vulnerable to energetic deficiencies [31].

Therefore, in order to better understand and treat glaucoma, it is pivotal to understand the retinal energetic metabolism—in particular, the metabolism of the RGC [30].

Essential factors in this complex energy metabolism are the rate of blood flow, the oxygen supply, the rate of glucose usage, and the function of the mitochondria [30].

Retinal mitochondria, which are involved in the axonal energetic support, become defective in glaucomatous models. There is a direct relationship between mitochondrial dysfunction and the elevation of intraocular pressure [32].

The vascular disturbance acts as a trigger for the glaucomatous damage and induces ischemia and reperfusion, thus enhancing oxidative stress (Figure 2). During an ischemic event, there is a pathological increase in neurotransmitters such as glutamate, which stimulates N-methyl-D-aspartate (NMDA) receptors to allow an intracellular influx of calcium, subsequently activating a calcium-dependent signaling cascade. This process leads to an increase in oxygen levels and nitrogen derivative levels (nitric oxide, peroxynitrite, and superoxide). These free radicals increase the conductance of TRPM7 (transient receptor potential channel 7), increasing the calcium influx and the production of oxygen and nitrogen species even more, culminating in cell death [32,33,34].

Among individuals of older age and individuals with a family history of glaucoma, the retinal oxidative stress levels are elevated, causing local tissue stress and subsequent inflammation. Abnormal inflammatory signaling leads to astrocyte activation and extracellular matrix remodeling, causing trabecular meshwork blockages and subsequent elevated intraocular pressure. As astrocytes continue to be activated as a result of abnormal inflammatory signaling, they cause direct injury to retinal ganglion cells through a glutamate excitotoxic mechanism [35].

Although excitotoxicity is considered the primary mechanism of neuronal dysfunction and cell death, antagonists of glutamatergic receptors have been unsuccessful in clinical trials with patients suffering ischemia or stroke [35]. Martinez-Gil et al. focused on the transient receptor potential channel 7, and whether it could contribute to the dysfunction found in ischemic retinal pathologies [36].

They used an experimental model of acute retinal ischemia and analyzed the changes in retinal function and retinal morphology and showed that retinal ischemia induces Müller cell gliosis (which can include the release of proinflammatory cytokines such as TNF-α, an excess production of NO, and the triggering of direct or indirect cytotoxic effects), along with increased TRPM7 channels in Müller cells. This suggests that TRPM7 could arise as a potential therapeutic target to be explored in retinal pathologies associated with ischemia, such as glaucoma [36].

An important role in the degeneration of RGC is played by NADPH oxidase (NOX) 1 and NOX4, which are expressed within retinal cells, being involved in reactive oxygen species (ROS) accumulation resulting from oxidative stress. Taking into account this pathological process, Liao et al. investigated the efficacy of setanaxib (a potent and highly selective inhibitor of NOX1 and NOX4) in ameliorating retinal ischemia–reperfusion (I/R) injury and found that it effectively inhibited NOX1 and NOX4, thereby regulating ROS production and preventing the activation of apoptosis and senescence-related factors in RGCs, ultimately protecting them against retinal I/R injury. These results suggest that setanaxib has great potential as a therapeutic target in glaucoma [36,37].

Through oxidation, the molecular structure of albumin is modified. The reaction between oxidative molecules and albumin leads to the formation of ischemia-modified albumin (IMA), currently known as a novel ischemic and oxidative biomarker [38,39,40]. Rusmayani and colleagues analyzed the concentration of IMA in the aqueous humor and blood plasma, and correlated these values with the degree of RNFL thinning in patients with glaucoma [41]. They found that, compared to healthy subjects, glaucoma patients present higher aqueous humor IMA, therefore identified as a marker of local ischemia and oxidative stress. The heightened local IMA described the chronic ischemic process associated with long-standing high IOP in POAG eyes. Their study also revealed that systemic oxidative activity is not an accurate representation of local ocular oxidative stress [41].

In their research, they also identified an important relationship between IMA concentration in the aqueous humor and RNFL thinning. Therefore, it was postulated that, by following this marker of ischemia, RNFL damage may be identified even before it is identified through Optical Coherence Tomography (OCT) examinations. Taking into account these findings, it can be stated that novel biomarkers may prove significant in detecting the disease in its early stages before extensive structural damage appears [41].

## 4. Vascular Endothelial Growth Factors and Glaucoma

Vascular endothelial growth factors are intimately involved in ocular diseases, both as pathological promoters and as therapeutic targets. Angiogenesis is the process of forming new blood vessels from existing ones, and it is essential in many physiological functions of the body. While it normally occurs in ocular tissues—the retina, choroid, or peripheral cornea—it also relates to pathologies such as age-related macular degeneration, retinal vessel occlusions, diabetic or prematurity retinopathies (prematurity, which may, in itself, be associated with the risk of congenital glaucoma [42]), and also abnormal neovascularization of the cornea [43]. Hypoxia is a potent up-regulator of angiogenesis factors, mainly vascular endothelial growth factors (VEGF) [44]; therefore, these factors may be involved in the pathogenesis of glaucoma. As we have stated earlier, ischemia plays a significant role.

Histological analysis reveals that VEGF-C is up-regulated in tissues involved in glaucoma pathogenesis, such as the trabecular meshwork, Schlemm’s canal, and aqueous humor collecting ducts, in a patient with POAG who underwent trabeculectomy, and also in a patient with neovascular glaucoma (NVG) secondary to proliferative diabetic retinopathy [45]. Interestingly, the same study found that VEGF-C is found in significantly lower concentrations in the aqueous humor of POAG and NVG patients compared to normal controls. The authors suggest this finding may be explained either by the consumption of VEGF-C by the trabecular meshwork cells, which overly express VEGF receptors, or by the loss of cells producing VEGF following the chronic stress of the disease [45].

While the ischemic theory seeks to describe the pathophysiology of open-angle glaucoma, ischemia is a process potentially involved in angle-closure glaucoma (PACG) as well. A study compared iris histological pieces collected through the trabeculectomy surgery from either POAG cases or PACG cases. Genetic analysis revealed that PACG irises expressed in a significantly higher proportion the genes responsible for coding vascular endothelial growth factors B and C (VEGFB, VEGFC) and the VEGF receptor 2 [46,47]. This overexpression of VEGF genes may reflect the tendency of a PACG eye to undergo ischemia [43,44].

Other angiogenic factors have been involved in the pathogenesis of neovascularization besides VEGF, with future research and therapeutic potential in glaucoma and neovascular ocular disease. In the pathway activated by the binding of VEGF to its receptor, several molecular cascades are stimulated, including the RAS, PI3K (phosphatidylinositol 3-kinase), and ERK/MAPK (extracellular signal-regulated kinases/mitogen-activated protein kinase). Moreover, these molecular pathways may signal to the stimulation of VEGF production, through hypoxia-inducible factor 1 (HIF-1)-dependent and independent mechanisms [48]. In different cellular models, RAS pathways have also been shown to induce both VEGF and other angiogenic factors (placenta growth factor - PGF, angiopoietin-2) [48]. In terms of translational research, treatment of animal model glaucoma with D-amino acid oxidase (DAAO) has been shown to increase phosphorylation of ERK1/2 and increase the expression of p-MEK, an upstream regulator of ERK; these alterations are identified mainly in the ganglion cell layer and Müller cells [49]. 

## 5. Neovascular Glaucoma and Normal-Tension Glaucoma

Two subtypes of glaucoma in which ischemia is more globally involved are neovascular glaucoma (NVG) and normal tension glaucoma (NTG).

Neovascular glaucoma is a secondary ocular pathological condition associated with numerous systemic diseases and ocular conditions and almost all of them share a common mediator of retinal ischemia and hypoxia. Diabetic retinopathy, ischemic central retinal vein occlusion, and ocular ischemic syndrome are the most common predisposing conditions for NVG [50].

Retinal ischemia stimulates angiogenesis, as it disrupts the balance between pro-angiogenic (including hepatocyte growth factor, VEGF, insulin-like growth factor, TNF, and IL 6) and anti-angiogenic factors (such as thrombospondin, somatostatin, pigment epithelium-derived factors, or transforming growth factor-beta) [51].

The major inciting factor implicated in the pathophysiology is VEGF, which is produced by a variety of cells in the retina like Muller cells, retinal pigment epithelium, pericytes, and ganglion cells, as well as by the non-pigmented ciliary epithelium [52]. VEGF stimulates endothelial cell migration as well as leukocyte adhesion, leading to a breakdown of the blood–retinal barrier. TGF-beta induces the formation of the fibroblast proliferation and the formation of the fibrovascular membrane. Consequently, it results in the formation of new leaky vessels in the anterior chamber, which obstruct the trabecular meshwork. This leads to a rise in IOP and can rapidly progress to glaucomatous optic neuropathy [53].

It is of note that low arterial pressure may increase retinal and optic nerve ischemia, as it leads to an imbalance in ocular perfusion pressure (the difference between systemic blood pressure and intraocular pressure). Therefore, patients with NTG should be instructed to avoid low arterial blood pressure [54,55].

NTG is an optic nerve neuropathy that consists of a variety of pathophysiological concepts. The primary mechanism in the development of NTG could be vascular dysfunction and consequently impaired autoregulation in response to low ocular perfusion pressure creating mild and repeated ischemic insults [56].

Vascular dysregulation involves the inability of a tissue to maintain a constant blood supply despite changes in perfusion pressure. Therefore, an insufficient blood supply to the optic nerve will induce ischemic and hypoxic processes in the RGCs and their axons, contributing to the development of glaucomatous pathology. Further, vascular abnormalities such as vasospasm syndrome and migraines were reported as risk factors for optic nerve damage in NTG [56].

Taking into consideration that lowering the IOP does not halt the disease progression in all cases also shows that additional mechanisms induce RGC death. Some NTG patients suffer from Alzheimer’s disease, suggesting that this neuropathy is an early manifestation of a more generalized neurodegenerative disease [57].

## 6. Ocular Blood Flow Measurement—A Diagnostic and Monitoring Approach in Glaucoma

The main functions of ocular blood flow are adequate oxygen and nutrient supply to the eye. Ocular blood flow (OBF) related to glaucoma is an area of study with increasing interest. Although raised intraocular pressure is the most important risk factor for the development and progression of glaucoma, in glaucoma eyes, the OBF is lower [58].

As previously mentioned, ocular blood flow is important in the pathophysiology of glaucoma. In patients with glaucoma, blood flow abnormalities are seen in virtually all parts of the eye, including the iris, retina, and choroid, but the blood perfusion deficit is particularly pronounced in the region of the vessels supplying the optic nerve. Ocular blood flow is more impaired in patients with progressive glaucoma than stable glaucoma [59]. The studies described that blood flow parameters correlate with visual field defects [60].

Ocular blood flow measurement is done using several techniques, but there is no “gold-standard” technique to date.

Currently, Color Doppler Imaging is the method of choice to measure blood flow parameters in retrobulbar circulation. Numerous studies have shown that there is an association between decreased blood flow velocities in retrobulbar circulation and glaucomatous damage. Also, Color Doppler Imaging is a valuable tool not only for diagnosing but also monitoring glaucoma patients [61].

Taking into consideration that retinal ganglion cells are supplied by retinal circulation and that the RGCs are lost in glaucoma, retinal blood flow is of great importance in understanding the pathophysiology of glaucoma [60].

The technologies used to assess retinal blood alterations include Laser Doppler Flowmetry, digital scanning laser angiography, laser speckle flowgraphy, Fourier domain optical coherence tomography, and retinal oximetry, which differ in terms of the parameters they use for the calculation of ocular circulation. All these technologies demonstrated a correlation between retinal blood flow and visual function. Some of these current imaging technologies can also measure the retinal morphologic changes that include thinning of the RNFL and excavation of the optic nerve head, which is a consequence of pathological loss of retinal ganglion cells in glaucoma [60].

Traditionally, imaging of the optic nerve head has proven to be difficult because of the complex vascularization. Recently, OCT Angiography (OCTA) was proven to generate fast angiographic images of the optic nerve head and retina. In comparison to traditional OCT, OCTA generates a picture of blood flow at a specific point in time and maps the vasculature. OCTA also shows vascular details, as dye techniques do, although it does not show slow vessel leaks visualized using dye angiography [61].

Optic nerve head alterations in glaucoma include reduced blood flow to the peripapillary retina and neuroretinal rim. In addition, blood flow at the neuroretinal rim corresponds to regional visual field defects in patients with normal tension glaucoma and primary open angle glaucoma. In patients with glaucoma, Scanning laser ophthalmoscopy (SLO) angiography revealed leakage from optic nerve head capillaries, suggesting peripapillary ischemia. Besides RNFL thinning, some studies suggested that ocular blood flow is associated with optic disk changes [62].

The choroidal blood flow can be measured with few imaging technologies, including SLO angiography, dynamic contour tonometry, and laser interferometric measurements of fundus pulsation—SLO angiography being the only one that measures it directly [60].

In addition to changes in blood flow parameters described in glaucoma, morphologic changes in the retina, optic nerve head, and choroid are associated with altered blood flow. These findings promote the possibility of ischemic injury in glaucoma.

## 7. The Relationship Between Ischemia, Glaucoma, and Myopia

Myopia is one of the most significant ocular pathologies, both through the high incidence—half of the world’s population is estimated to be myopic by the year 2050—and through the potential comorbidities it entails [63]. It is well known that myopia is a significant risk factor for the development of glaucoma; however, the morphological changes that appear in both glaucoma and myopia are still a matter of debate. Oxidative stress has been discovered as a pathological factor in both myopia and glaucoma [64]; more importantly, vascular changes have been reported in both pathologies.

The studies revealed that both glaucomatous and myopic eyes have parallel vascular changes, such as retinal microvasculature attenuation, decreased capillary density, reduced retinal, choriocapillaris, and optic nerve head (ONH) blood flow. Also, the comparison of vascular features of glaucomatous patients with and without myopia showed that myopic glaucomatous eyes presented greater vascular changes than non-myopic glaucomatous eyes, which included larger reductions in choroidal blood flow and velocity, lower macular and peripapillary capillary density and impaired peripapillary vasoreactivity. Hypotheses that the relationship between myopia and glaucoma might be vascular in nature, and may be present before the glaucomatous degeneration, are beginning to escalate in the research domain [65,66,67]. 

## 8. Potential Therapeutic Approaches to Ischemia in Glaucoma

As an important element of glaucoma physiopathology, ischemia may prove a valuable therapeutic target. Currently, the only therapeutic target is the IOP, through either medical treatment (systemic or more commonly topical), laser intervention, or surgical procedures [68]. The first line of treatment, the topical medications, act by either increasing the outflow of aqueous humor through the uveoscleral pathway (prostaglandin analogs), by decreasing the humor production (carbonic anhydrase inhibitors, beta-blockers), by both mechanisms (alpha-adrenergic agonists), or, more recently, by increasing the outflow through the conventional trabecular pathway (rho kinase inhibitors) [68].

One recently approved medication may prove valuable against the ischemic phenomena in glaucoma: latanoprostene bunod. This molecule combines a prostaglandin F2α analog (with the above mentioned mechanism in IOP lowering) with nitric oxide (NO)-donating moiety—which is quickly freed when reaching the ocular surface [69]—in contact with the cornea, the NO is enzymatically released and increases aqueous humor outflow through the conventional pathway by relaxing trabecular meshwork cells and cells in the walls of the Schlemm’s canal [70].

It is known that NO is a widely found signaling molecule in the human body, with a particular role in modulating vascular tone (by vasodilation); therefore, its production is upregulated in ischemic conditions [71]. In the eye, NO has been described as an endogen molecule in the retina, choroid, trabecular meshwork, and ciliary body [71], and most importantly, in the optic nerve head, where it modulates blood flow and oxygenation [72]. It has been reported that a NO synthase inhibitor, in experimental conditions, led to the vasoconstriction of retinal vessels, decreased choroid and retinal blood flow, and, consequently, IOP increase [73], further supporting the role of NO in adequate ocular perfusion and modulation of IOP and IOP-led glaucomatous damage.

More research in humans is needed to fully describe the ischemia protection role of NO in glaucoma patients; however, the first data seems promising. A recent study involving ocular hypertension (OHT) and open-angle glaucoma patients (OAG), in comparison with normal subjects, followed macular blood vessel density after 4 weeks of topical administration of latanoprostene bunod. OCTA revealed that, compared to the baseline, whole macular vessel density improved significantly after latanoprostene bunod administration, in OHT and OAG patients, but not in normal subjects. While vessel density in the peripapillary area was not improved, this study suggests that latanoprostene bunod may play a significant role in retinal vascular regulation, especially in eyes affected by glaucoma or high IOP [74].

Several molecules are in the preclinical to clinical studies pipeline, and may address the ischemic component of the disease. As mentioned above, Endothelin-1 is involved in the pathogenesis of glaucoma. A recent meta-analysis has shown that, compared to healthy controls, glaucoma patients have a significantly higher ET-1 concentration in plasma, and this includes multiple types of glaucoma—primary open angle, angle closure, and normal tension glaucoma. Moreover, aqueous humor concentrations are significantly higher in glaucoma, particularly in open-angle cases [75]. NCX 434 is a molecule composed of the glucocorticoid triamcinolone acetonide (TA) and NO, and was studied in animal models of ocular ischemia, which was performed by administering Endothelin-1. It was shown that the NO part of the molecule has protected against IOP rise (which may be secondary to the TA) and against ET-1-induced vasoconstriction (assessed through echography of the ophthalmic artery and electroretinography) [76].

A complex research paper has recently investigated the ongoing clinical trials for glaucoma. As expected, 92% of investigated molecules act on IOP lowering, but 3% of them act on the vascular component (choroidal, retinal, optic disc, or retrobulbar blood flow) [77]. Phosphodiesterase inhibitors, which act by increasing blood circulation, have been involved in a clinical trial for glaucoma [77]; however, other clinical and preclinical studies may suggest a deleterious effect on IOP [78]. That study has been suspended due to COVID-19 pandemic restrictions [79].

One other promising vascular target in glaucoma is cannabinoid receptors [77]. Cannabinoid receptors 1 and 2 can be found in the eye in the retina and retinal pigmentary epithelium, ciliary body and iris, trabecular meshwork, and Schlemm’s canal [80]. It has been reported that potential cannabinoid benefits in glaucoma may include vasodilation in the ciliary muscle (which may decrease aqueous humor production), neuroprotection, inhibition of reactive oxygen species production, and consequently, an IOP decrease [81]. One clinical study is currently underway, in which dronabinol, a synthetic Tetrahydrocannabinol agent, is compared to a placebo in glaucoma and healthy subjects in terms of optic nerve head blood flow, retinal vessel diameter, blood flow, and oxygen saturation [82].

## 9. Conclusions and Future Directions

To summarize, the pathogenesis of glaucoma is complex and is still a matter of research. The vascular theory explains glaucomatous neuropathy as a consequence of the interplay between the increased IOP, oxidative stress, and vascular factors, which leads to tissue ischemia. Further proof of vascular importance in glaucoma is normotensive glaucoma, a subtype in which high IOP is not as significant, but the main risk factor is low ocular perfusion pressure.

As the retina is a highly metabolically active tissue, deficiencies in blood flow will severely damage the energy metabolism and contribute to oxidative damage. Moreover, as hypoxia becomes an important stimulus in glaucoma, vascular endothelial growth factors and their receptors are upregulated and significantly expressed, increasing the risk for ocular neoangionegesis.

As a result of the impact of the vascular theory, both paraclinical investigations and therapeutic approaches are in development. Currently, Color Doppler Imaging for investigating the retrobulbar circulation OCT Angiography for the optic nerve head is the standard option, but others are in development and testing for glaucoma patients.

Not last, new therapies targeting ocular ischemia may prove valuable for glaucoma patients. The most advanced one is latanoprostene bunod, which is available on the market, combining the benefits of prostaglandin analogs and the vasodilation of nitric oxide.

Glaucoma patients require lifelong treatment and follow-up, and the disease has a significant negative impact on patients’ quality of life in terms of anxiety, psychological well-being, daily life, driving, and confidence in healthcare. We believe that researching and developing the ischemic pathway in glaucoma will open new and exciting opportunities for our patients’ well-being.

## Figures and Tables

**Figure 1 ijms-25-12400-f001:**
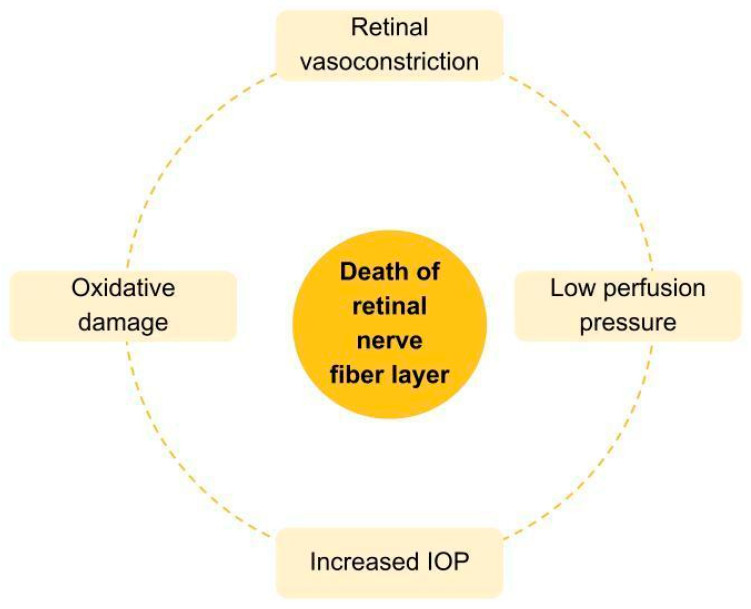
A model of vascular theory in glaucoma pathogenesis. The interplay between the increased IOP, oxidative stress, and vascular factors that lead to tissue ischemia is responsible for the damage in the retinal nerve fiber layer and, subsequently, glaucomatous neuropathy.

**Figure 2 ijms-25-12400-f002:**
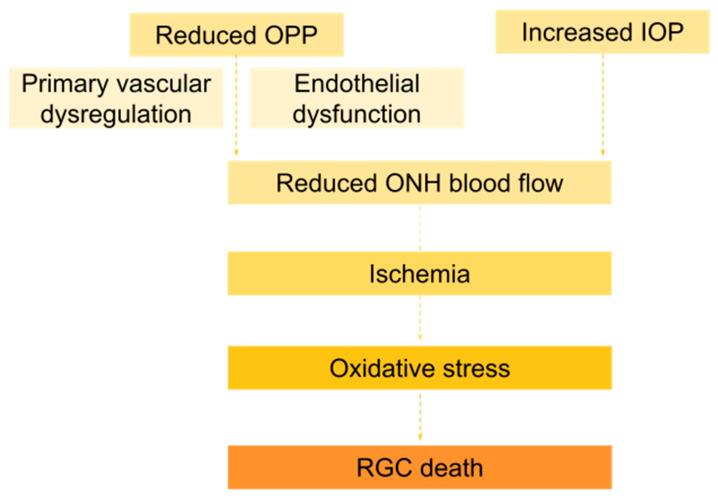
Suggested mechanism involving ischemia, oxidative stress, and, ultimately, retinal ganglion cell death, which is the key pathogenic mechanism in glaucoma.

## Data Availability

No new data were created.

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
