# Peer review of "Interplay Between Ocular Ischemia and Glaucoma: An Update"

_ijms, 2024, doi:10.3390/ijms252212400_

Round 1

Reviewer 1 Report

Comments and Suggestions for Authors

This paper provides a comprehensive review of the relationship between ischemia and glaucoma. The authors discuss key ischemic mechanisms, including the roles of vascular endothelial dysfunction and oxidative stress in glaucoma pathogenesis. They also explore techniques for monitoring ocular blood flow and emerging therapeutic targets aimed at modulating ischemia. Overall, the paper offers valuable insights into glaucoma's mechanism in the aspect of ischemia. However, the overall flow of the paper feels somewhat disjointed, and specific sections would benefit from reorganization to improve readability and coherence.

Comments:

1. Please show the full name of Primary Open-Angle Glaucoma (POAG) upon the first appearance.

2. Figure 1 is a bit unclear. How does the hypotension or hypertension influence the ocular perfusion? What is the role of IOP in the circle of hypoperfusion, oxidative damage, and vasoconstriction? Please modify it to better align with the discussion in the main text.

3. When discussing the ischemia theory, I think it is worth noting that the retina is one of the highest oxygen consuming and the most metabolically active tissues in the body, which requires efficient mechanism to maintain homeostasis and makes it highly sensitive to ischemia conditions.

4. The section on ocular blood flow measurement (Section 6) would benefit from restructuring to improve clarity. A more logical flow might begin with an overview of the importance of measuring ocular blood flow, followed by a summary of all available techniques. Afterward, a detailed discussion of each technique (e.g., Color Doppler, OCTA, SLO) could be provided. Currently, the section jumps between topics, making it difficult to follow the narrative.

5. The conclusion feels abrupt and lacks cohesion. It jumps between ideas without fully synthesizing the review's key points. Consider reorganizing this section to better summarize the paper's findings, emphasize the significance of ischemia in glaucoma, and suggest clear future directions for research or clinical applications.

Author Response

Thank you so much for taking the time to read our work and suggest improvements! Please find attached a point-by-point explanation and response, and the manuscript in which we have highlighted modifications in green. 

Review 1 green

  • Please show the full name of Primary Open-Angle Glaucoma (POAG) upon the first appearance. 

Thank you for this indication, please find line 97:

There are studies that showed elevated basal nitric oxide synthase (NOS) activity in Primary open-angle glaucoma (POAG) suggesting a compensatory mechanism [...]

  • Figure 1 is a bit unclear. How does hypotension or hypertension influence the ocular perfusion? What is the role of IOP in the circle of hypoperfusion, oxidative damage, and vasoconstriction? Please modify it to better align with the discussion in the main text.

Thank you, we have remade the figure and detailed the caption to clearly illustrate the factors implicated in the vascular theory.

  • When discussing the ischemia theory, I think it is worth noting that the retina is one of the highest oxygen consuming and the most metabolically active tissues in the body, which requires an efficient mechanism to maintain homeostasis and makes it highly sensitive to ischemia conditions.

Thank you for this suggestion, we have added the following discussion, please find lines 135-153.

The retina is one of the most metabolically active tissues in the body and has the highest oxygen demand. It is estimated that, in humans, the oxygen extraction is approximately 2.33 μl of oxygen per minute, or 1.42 ml per minute per 100 g of retinal tissue, higher than other tissues [...]. As energy consumption of the retina is of a similar value to that of the brain and the retinal vasculature is very sensitive to oxygen availability, it requires an efficient mechanism to maintain homeostasis [...]. As such, the normal functioning of retinal neurons can be affected if nutrients and oxygen are not supplied according to the high tissular demands [...].

As retinal ganglion cells are rich in mitochondria, they have an exceedingly active metabolism and are particularly vulnerable to energetic deficiencies.[...].

Therefore, in order to better understand and treat glaucoma, it is pivotal to understand the retinal energetic metabolism, in particular the metabolism of the RGC.[...].

Essential factors in this complex energy metabolism are the rate of blood flow, the oxygen supply, the rate of glucose usage and the function of the mitochondria. (energy metabolism in the inner retina in health and glaucoma).

  • The section on ocular blood flow measurement (Section 6) would benefit from restructuring to improve clarity. A more logical flow might begin with an overview of the importance of measuring ocular blood flow, followed by a summary of all available techniques. Afterward, a detailed discussion of each technique (e.g., Color Doppler, OCTA, SLO) could be provided. Currently, the section jumps between topics, making it difficult to follow the narrative.

We have clarified the section, please find lines 270-282.

  • The conclusion feels abrupt and lacks cohesion. It jumps between ideas without fully synthesizing the review's key points. Consider reorganizing this section to better summarize the paper's findings, emphasize the significance of ischemia in glaucoma, and suggest clear future directions for research or clinical applications.

Thank you, we have completely rewritten the conclusions to better reflect the main ideas from each section of the paper:

To summarize, the pathogenesis of glaucoma is complex and still a matter of research. The vascular theory explains the glaucomatous neuropathy as a consequence of the interplay between the increased IOP, oxidative stress and vascular factors which lead to tissue ischemia. Further proof of the vascular importance in glaucoma is the normotensive glaucoma, a subtype in which high IOP is not as significant, but the main risk factor is the low ocular perfusion pressure.

As the retina is a highly metabolically active tissue, deficiencies in blood flow will severely damage the energy metabolism and contribute to oxidative damage. Moreover, as hypoxia becomes an important stimulus in glaucoma, vascular endothelial growth factors and their receptors are upregulated and significantly expressed, increasing the risk for ocular neoangionegesis. 

As a result of the impact of the vascular theory, both paraclinical investigations and therapeutic approaches are in development. Currently, Color Doppler Imaging for investigating the retrobulbar circulation OCT Angiography for the optic nerve head are standard options, but others are in development and testing for glaucoma patients.

Not last, new therapies targeting ocular ischemia may prove valuable for glaucoma patients. The most advanced one is latanoprostene bunod, available on the market, combining the benefits of prostaglandin analogues and the vasodilation of nitric oxide. 

Glaucoma patients require lifelong treatment and follow-up, and the disease has a significant negative impact on patients’ quality of life in terms of anxiety, psychological well-being, daily life, driving and confidence in healthcare. We believe that researching and developing the ischemic pathway in glaucoma will open new and exciting opportunities for our patients’ wellbeing.

We hope that these revisions are sufficient to make our manuscript suitable for publication in the International Journal of Molecular Sciences Journal. We are looking forward to hearing from you at your earliest convenience.

Sincerely,

Miruna Burcel MD

2 Nov 2024

Reviewer 2 Report

Comments and Suggestions for Authors

The article provides a comprehensive review of the role of ischemia and oxidative stress in the pathogenesis of glaucoma. It discusses how vascular dysregulation and oxidative damage lead to the death of retinal ganglion cells, emphasizing the importance of ischemia as a risk factor for glaucoma and a potential therapeutic target.|

Advantages:

1. Comprehensive coverage: The manuscript reviews the mechanical and vascular theories of glaucoma and cites relevant studies supporting the involvement of ischemia in the progression of the disease.

2. Novel biomarkers: The authors highlight potential early biomarkers for glaucoma, providing value for early diagnosis.

3. New therapeutic approaches: New therapies such as latanoprostene bunod and NO modulation are being focused on to address the intraocular pressure (IOP) and ischemic factors.

Weaknesses and suggestions:

This article could better integrate multiple theories (ischemic, mechanical, and oxidative stress) into a model of glaucoma progression. A more cohesive theoretical integration would provide clearer insights into the pathogenesis of glaucoma.

Overall, this manuscript provides valuable insights into the mechanisms of ischemia and oxidative stress in glaucoma, highlighting promising therapeutic targets. Although the article is comprehensive, it could be strengthened by better integration of various other pathophysiological theories. The study contributes to the understanding and treatment of glaucoma beyond IOP management.

Comments on the Quality of English Language

 Nessuna comunicazione

Author Response

Thank you for your comments, but please write the English version. 

Round 2

Reviewer 1 Report

Comments and Suggestions for Authors

The revised version has addressed my previous comments; I don't have further comments.

Author Response

1) The Myopia-Glaucoma-Ocular ischemia section is missing. Short descriptions with recent findings are recommended.

Myopia is one of the most significant ocular pathologies, both through the high incidence - half of the world’s population is estimated to be myopic by the year 2050, and through the potential comorbidities it entails [63]. It is well known that myopia is a significant risk factor for the development of glaucoma, however the morphological changes that appear in both glaucoma and myopia are still a matter of debate. Oxidative stress has been discovered as a pathological factor in both myopia and in glaucoma [64], and more importantly, vascular changes have been reported in both pathologies.

The studies revealed that both glaucomatous and myopic eyes have parallel vascular changes such as retinal microvasculature attenuation, decreased capillary density and reduced retinal, choriocapillaris and ONH blood flow. Also, the comparison of vascular features of glaucomatous patients with and without myopia, showed that myopic glaucomatous eyes presented greater vascular changes that non-myopic glaucomatous eyes, which included larger reductions in choroidal blood flow and velocity, lower macular and peripapillary capillary density and impaired peripapillary vasoreactivity. Hypotheses that the relationship between myopia and glaucoma might be vascular in nature, and may be present before the glaucomatous degeneration, are beginning to escalate in the research domain [65,66][67].

2) Hypoxia and molecular pathway parts are less discussed. Not only VEGF but also other pathologic angiogenic factors including upper stream transcription factors such as ERK1/2, C/EBP β, c-Fos or HIF are recommended to be listed in the relevant section.

Other angiogenic factors  have been involved in the pathogenesis of neovascularization besides VEGF, with future research and therapeutic potential in glaucoma and neovascular ocular disease. In the pathway activated by the binding of VEGF to its receptor, several molecular cascades are stimulated, including the RAS, PI3K (phosphatidylinositol 3-kinase), ERK/MAPK (extracellular signal-regulated kinases/mitogen-activated protein kinase). Moreover, these molecular pathways may signal to the stimulation of VEGF production, through hypoxia-inducible factor 1 (HIF-1) dependent and independent mechanisms [48]. In different cellular models, RAS pathways have also been shown to induce both VEGF and other angiogenic factors (placenta growth factor - PGF, angiopoietin-2) [48]. In terms of translational research, treatment of animal model glaucoma with D-amino acid oxidase (DAAO) has been shown to increase phosphorylation of ERK1/2 and increase the expression of p-MEK, an upstream regulator of ERK - these alterations being identified mainly in the ganglion cell layer and Müller cells [49].